# Multifunctional Hybrid Nanozymes for Magnetic Enrichment and Bioelectrocatalytic Sensing of Circulating Tumor RNA during Minimal Residual Disease Monitoring

**Kevin M. Koo** [1,2]

1    The University of Queensland Centre for Clinical Research (UQCCR), Brisbane, QLD 4029, Australia; maisheng.koo@uqconnect.edu.au
2    XING Applied Research & Assay Development (XARAD) Division, XING Technologies Pty. Ltd., Brisbane, QLD 4073, Australia

**Abstract:** Iron oxide nanozymes are a form of nanomaterial with both superparamagnetic and enzyme-mimicking properties. Ongoing research efforts have been made to create multifunctional iron oxide hybrid nanozymes with auxiliary properties through biomolecular modifications. Such iron oxide hybrid nanozymes can be useful for rapid and cost-effective analysis of circulating tumor nucleic acids (ctNAs) in patient liquid biopsies during minimal residual disease (MRD) monitoring of cancer recurrence. Herein, the use of streptavidin-modified iron oxide hybrid nanozymes is reported for magnetic enrichment and bioelectrocatalytic sensing of three prostate cancer (PCa) ctRNA biomarkers with high detection specificity and sensitivity (10 copies) over an ultrabroad dynamic range (five orders of magnitude). Furthermore, the feasibility of ctRNA analysis for pre- and post-cancer treatment MRD monitoring is demonstrated using PCa urinary liquid biopsy samples.

**Keywords:** nanozymes; circulating tumor nucleic acid; liquid biopsy; magnetic enrichment; minimal residual disease; cell-free RNA





## 1. Introduction

Circulating tumor nucleic acids (ctNAs) are generally tumor-derived NA fragments that have been released into body fluids such as blood or urine via passive tumor cell death or active tumor cell secretion mechanisms [1,2]. As an alternative to traditional tissue biopsy procedures, ctNA detection offers unique liquid biopsy benefits of being minimally-invasive, easily obtainable and able to provide a "complete" sampling of cancer mutations present in different parts of the tumor [3–5]. In terms of clinical applications, targeted detection of cancer-specific ctNA biomarkers has indicated the prospect of minimal residual disease (MRD) monitoring [6–10]. MRD monitoring is the continuous surveillance of cancer recurrence following primary treatment by using liquid biopsy samples to detect trace levels of ctNAs, in order to enable early management of any recurrent disease.

For effective MRD monitoring, it is paramount to have technologies that are able to accurately discriminate trace copies of ctNA biomarkers from a background of excess wild-type NA sequences [11–13]. The most common workflow for enabling MRD monitoring is spin-column-based ctNA enrichment followed by next-generation sequencing (NGS) or polymerase chain reaction (PCR) technologies for ctNA detection. However, spin-column-based ctNA enrichment is time-consuming, laborious and may result in insufficient recovery and/or biased fragment size enrichment; and NGS/PCR detection technologies generally rely on the use of fluorescence labels and sophisticated readout instruments, which increases assay cost and time. Hence, ctNA detection for MRD monitoring can benefit from a strategy to rapidly enrich and detect ctNA biomarkers in a cost-effective manner.

Nanozymes are nanomaterials with intrinsic enzyme-like activities and have been widely used as non-biological enzyme mimetics for biomedical applications [14–19]. Iron oxide nanoparticles have been demonstrated to be peroxidase-mimicking nanozymes [20–22]

that can be utilized for electrochemical biosensing [23–25]. Additionally, iron oxide nanoparticles exhibit superparamagnetic behavior at the nanoscale and are useful for magnetic enrichment of biomolecules [26–30]. Our laboratory has recently conceived a bioelectrocatalytic cycling approach that exploits magnetics to concentrate DNA as a scaffold for bringing redox-active labels into direct electrode contact for superior electrochemical signaling [31,32]. To realize rapid enrichment and cost-effective detection of ctNA at an enhanced detection sensitivity during MRD monitoring, it is envisioned that a biomolecule-coupled iron oxide hybrid nanozyme can provide both magnetic enrichment and bioelectrocatalytic sensing of ctNA biomarkers from liquid biopsy samples.

Herein, the use of streptavidin-modified iron oxide hybrid nanozymes is reported for magnetic enrichment and the electroanalysis of multiple ctRNA biomarkers in urinary liquid biopsy samples during MRD monitoring. Using prostate cancer (PCa) as a disease model for pre- and post-cancer treatment MRD monitoring, three PCa ctRNA urinary biomarkers [33–35] are magnetically enriched and subjected to a unique bioelectrocatalytic system of NA-intercalated methylene blue (MB) redox labels and streptavidin-modified iron oxide nanozymes. Furthermore, the assay is demonstrated for cost-effective usage on disposable screen-printed electrodes with zmol (10 copies) detection sensitivity and an ultrabroad dynamic range over five orders of magnitude.

## 2. Results and Discussion

### 2.1. Streptavidin-Modified Iron Oxide Nanozymes for ctRNA Magnetic Enrichment and Bioelectrocatalytic Sensing

The approach of using streptavidin-modified iron oxide nanozymes for both magnetic enrichment and bioelectrocatalytic sensing of urinary ctRNA is shown in Figure 1. The detection of highly PCa-specific ctRNA biomarkers in urine has been shown to be a feasible form of liquid biopsy [33]. Such ctRNA biomarkers include the gene fusion between the *TMPRSS2* and *ERG* genes (T2:ERG), and the overexpression of *PCA3* and *KLK2*.

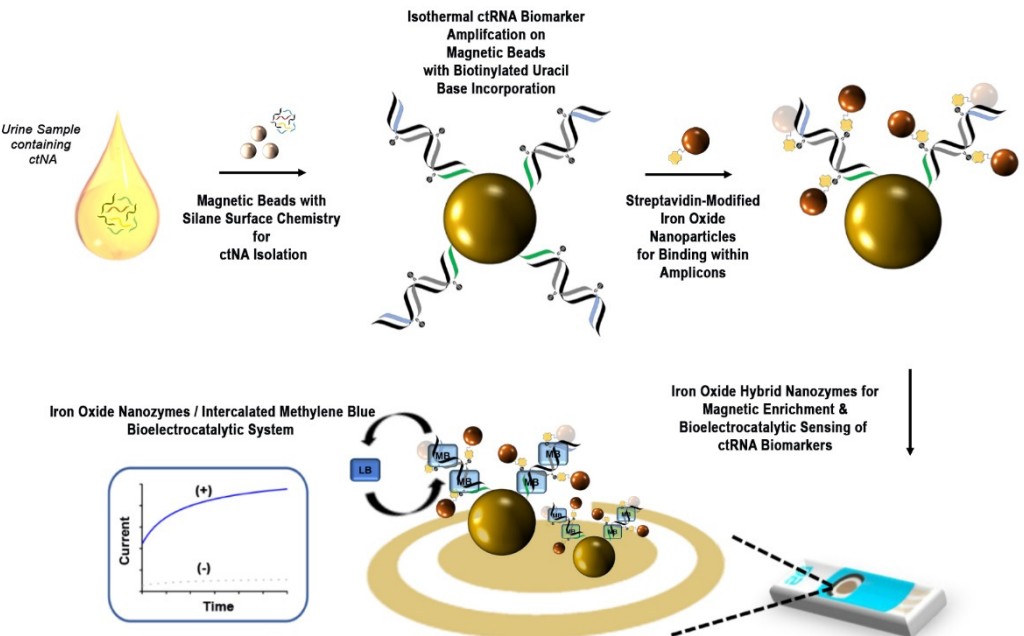

**Figure 1.** Schematic for using streptavidin-modified iron oxide nanozymes for urinary ctRNA magnetic enrichment and bioelectrocatalytic sensing.

The assay is first initiated by the direct addition of magnetic beads with silane surface chemistry to unprocessed urine samples. Under different salt conditions, ctNAs are magnetically isolated and released for rapid isothermal recombinase amplification of ctRNA biomarkers. Each biomarker is captured and amplified with biotinylated uracil bases

being incorporated into the resultant amplicons on primer-modified magnetic beads. After isothermal amplification, streptavidin-modified iron oxide nanoparticles are added for binding to the biotinylated uracil bases within the amplicons. The superparamagnetic property of the streptavidin-modified iron oxide nanoparticles is then utilized to magnetically enrich the ctRNA biomarker amplicons on independent screen-printed electrode surfaces for bioelectrocatalysis. Moreover, as compared to the micron-sized magnetic beads, the nanoscale of iron oxide nanoparticles is physically compatible for incorporation within similarly nanosized NA molecules by reducing steric hindrance. The use of magnetic concentration enables close contact of the amplicons to the electrode surface without prior time-consuming surface modification, as well as reducing electrochemical signal variability.

To commence bioelectrocatalytic sensing, NA-intercalating MB redox labels are first added to the ctRNA biomarker amplicons on the electrode surfaces. The amplicons serve as biological scaffolds for the intercalation of MB redox labels [31], and the streptavidin-modified iron oxide nanoparticles (incorporated within the amplicons) function as peroxidase-mimicking nanozymes to interact with MB for bioelectrocatalysis. During the bioelectrocatalytic process, intercalated MB is first reduced to leucomethylene blue (LB) with two freely diffusing electrons. The iron oxide nanozymes then serve as electron sinks near the electrode surface and return LB to oxidized MB to complete the bioelectrocatalytic process. Chronoamperometry is used to measure the current level of regenerated MB during bioelectrocatalytic sensing and the measured current is correlated to the ctRNA biomarker level. The signal enhancement due to bioelectrocatalytic cycling is evidently demonstrated (Figure S1) with a difference in the chronoamperometric measurements of methylene blue in the presence and absence of iron oxide nanozymes.

## 2.2. Optimization of Bioelectrocatalytic Signaling Process

The bioelectrocatalytic signaling process of our assay is dependent on the amount of ctRNA biomarker amplicons being magnetically enriched on the electrode surface. The ctRNA biomarker amplicons serve as the biological scaffolds for intercating MB redox labels and streptavidin-modified iron oxide nanozymes to generate enhanced electrochemical signaling. Thus, to achieve optimal signal enhancement, it is ideal to maximize the amount of ctRNA biomarker amplicons on the electrode surface and the signaling condition of the bioelectrocatalytic system. Therefore, parameters such as the isothermal amplification time (for producing more amplicons on the magnetic bead surface) and the pH of the electrolyte buffer (which is known to affect bioelectrocatalytic signaling kinetics) are optimized.

Firstly, different isothermal amplification times (5, 10, 15, 20 and 30 min) were tested for producing the optimal amount of ctRNA biomarker amplicons before bioelectrocatalytic sensing. It was found that a maximal electrochemical signal was observed at 20 min and no further increase in current response change was observed thereafter (Figure S1). This was probably due to maximal coverage of ctRNA biomarker amplicons on the magnetic bead surface at 20 min during magnetic enrichment on the electrode surface, thus leading to no signal improvement beyond this timepoint. Hence, we used a 20 min isothermal amplification time for all further experiments.

Next, the electrolyte buffer pH was optimized for achieving the maximal electrochemical signal. Using different buffer pH values between 5.0 and 10.0, it was found that the optimal pH for generating maximal electrochemical signal occurred at pH 7.5 (Figure S2). The current level being generated improved with increased pH up to pH 7.5 but subsequently decreased thereafter. Although it is widely known that iron oxide nanozymes generally exhibit enzymatic behavior under acidic conditions, recent studies have shown that it is feasible to achieve iron oxide nanozyme activity at physiological pH through surface modification with negatively charged molecules [36,37]. In this work, it is hypothesized that the negatively charged DNA amplicons (in which the hybrid iron oxide nanozymes were incorporated) provided a similar effect in stabilizing oxidized MB through electrostatic interactions for activity at pH 7.5. It is believed that the highest signal change was obtained at pH 7.5 due to electrode surface damage at lower pH levels; and

conversely, the ctRNA biomarker amplicons were prone to degradation at higher pH levels. Hence, the electrolyte buffer pH was optimized at pH 7.5.

### 2.3. Detection Specificity for Multiple ctRNA Biomarkers

Highly specific ctRNA biomarker detection is crucial for accurate discrimination from a high background of interfering biomolecules in liquid biopsy samples. Three ctRNA biomarker-specific (T2:ERG, *PCA3* and *KLK2*) primer pairs were designed to produce amplicons on independent sets of magnetic bead surfaces. To evaluate specific ctRNA biomarker amplification, each primer pair was tested on various well-characterized PCa cell lines (DuCap, LnCap, 22Rv1) with known expression of T2:ERG, *PCA3* and *KLK2*. Upon the addition of paired reverse primers to initiate isothermal amplification on corresponding forward-primer-modified magnetic beads, it was found that the three biomarker levels in respective PCa cell lines agreed with known expression levels (Figure 2) [23,38,39]. Standard quantitative polymerase chain reaction (qPCR) was also performed on contrived urine samples with different prostate cancer cell line RNA. The qPCR results (Table S2) were found to be 100% concordant. Thus, it was confirmed that our designed primer sets are highly specific for each of the three ctRNA biomarkers.

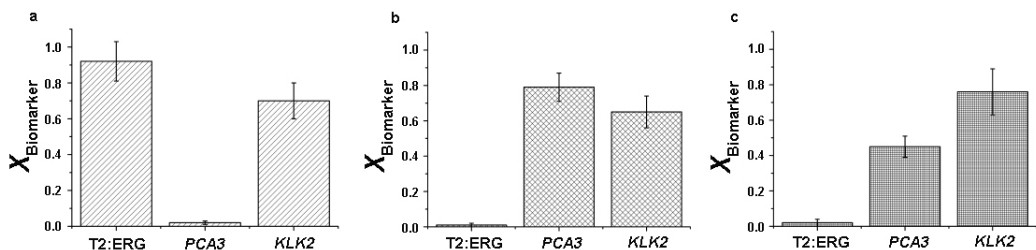

**Figure 2.** Detection of multiple ctRNA biomarkers in (**a**) DuCap, (**b**) LnCap, (**c**) 22Rv1 cell lines. Error bars represent standard deviations of three technical replicates.

### 2.4. Detection Sensitivity and Linear Dynamic Range

High detection sensitivity within a broad linear dynamic range is essential for trace ctRNA biomarker detection in liquid biopsy samples. To evaluate detection sensitivity and linear dynamic range, T2:ERG, *PCA3* and *KLK2* in vitro transcripts over a copy number range of 0–1,000,000 copies were evaluated for quantitative detection (Figure 3). It was observed that a clinically relevant detection sensitivity of 10 copies was consistent across the three biomarkers. The exceptional detection sensitivity was attributed to the combination of isothermal biomarker amplification, and the unique bioelectrocatalytic sensing of NA-intercalated MB redox labels and streptavidin-modified iron oxide nanozymes. In particular, the close proximity of iron oxide nanozymes and MB labels within the ctRNA biomarker amplicons is beneficial for significantly enhancing the resultant electrochemical signals. Furthermore, ultrabroad linear dynamic ranges of detection spanning over five orders of magnitude (0–100,000 copies) were obtained for each of the biomarkers. In addition, good day-to-day reproducibility of the assay was demonstrated with the three independent technical replicates.

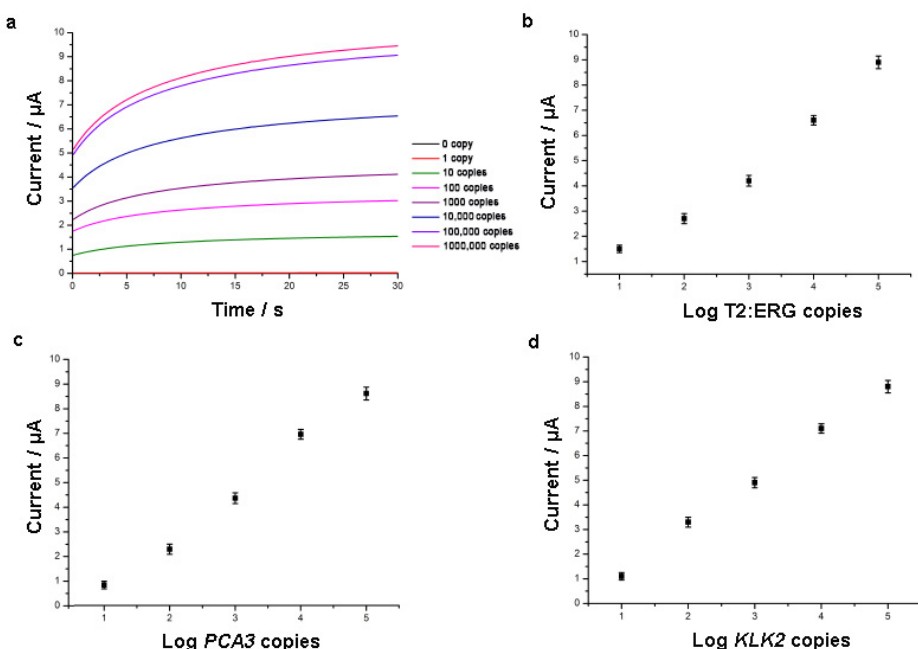

**Figure 3.** (**a**) Representative chronoamperometric signals for the detection of 0–1,000,000 T2:ERG copies. Detection sensitivity and linear dynamic range for (**b**) T2:ERG, (**c**) *PCA3*, (**d**) *KLK2*. Error bars represent standard deviations of three technical replicates.

In comparison to existing electrochemical assays for ctRNA detection in liquid biopsy samples [23,32,40–43], this work is distinct in utilizing streptavidin-modified iron oxide hybrid nanozymes for the dual function of magnetic enrichment and bioelectrocatalytic sensing of ctRNA biomarker amplicons. It is noteworthy that the use of streptavidin-modified iron oxide hybrid nanozymes resulted in a similar assay sensitivity performance as compared to the use of diffusive redox labels and qPCR [31], thus highlighting the effective concept of using streptavidin to bring peroxidase-mimicking nanozymes in close contact to MB and to the electrode surfacer for bioelectrocatalysis.

*2.5. Minimal Residual Disease Monitoring in Patient Urine Samples for Prostate Cancer Recurrence*

The use of urine for non-invasive MRD monitoring is beneficial for patients to be able to sequentially provide urine samples at minimal health risk and cost, especially for advanced PCa patients with non-accessible metastases such as bone-predominant lesions. To investigate the potential of avoiding serial invasive biopsies during longitudinal PCa recurrence monitoring, pre- and post-treatment urine samples of 10 PCa patients who had undergone radical prostatectomy treatment were used to perform ctRNA biomarker detection of T2:ERG, *PCA3* and *KLK2* (Figure 4). It was found that 4 of the 10 patients were identified with no significant decrease in T2:ERG, *PCA3* and *KLK2* levels after undergoing treatment, and these outcomes were correlated with diagnosed PCa biochemical recurrence. In contrast, the remaining patients displayed low gene expression of the ctRNA biomarkers and remained PCa recurrence free. These results highlighted the feasibility of using hybrid iron oxide nanozymes for PCa MRD monitoring from repeated liquid biopsies through the analysis of PCa-specific ctRNA biomarkers.

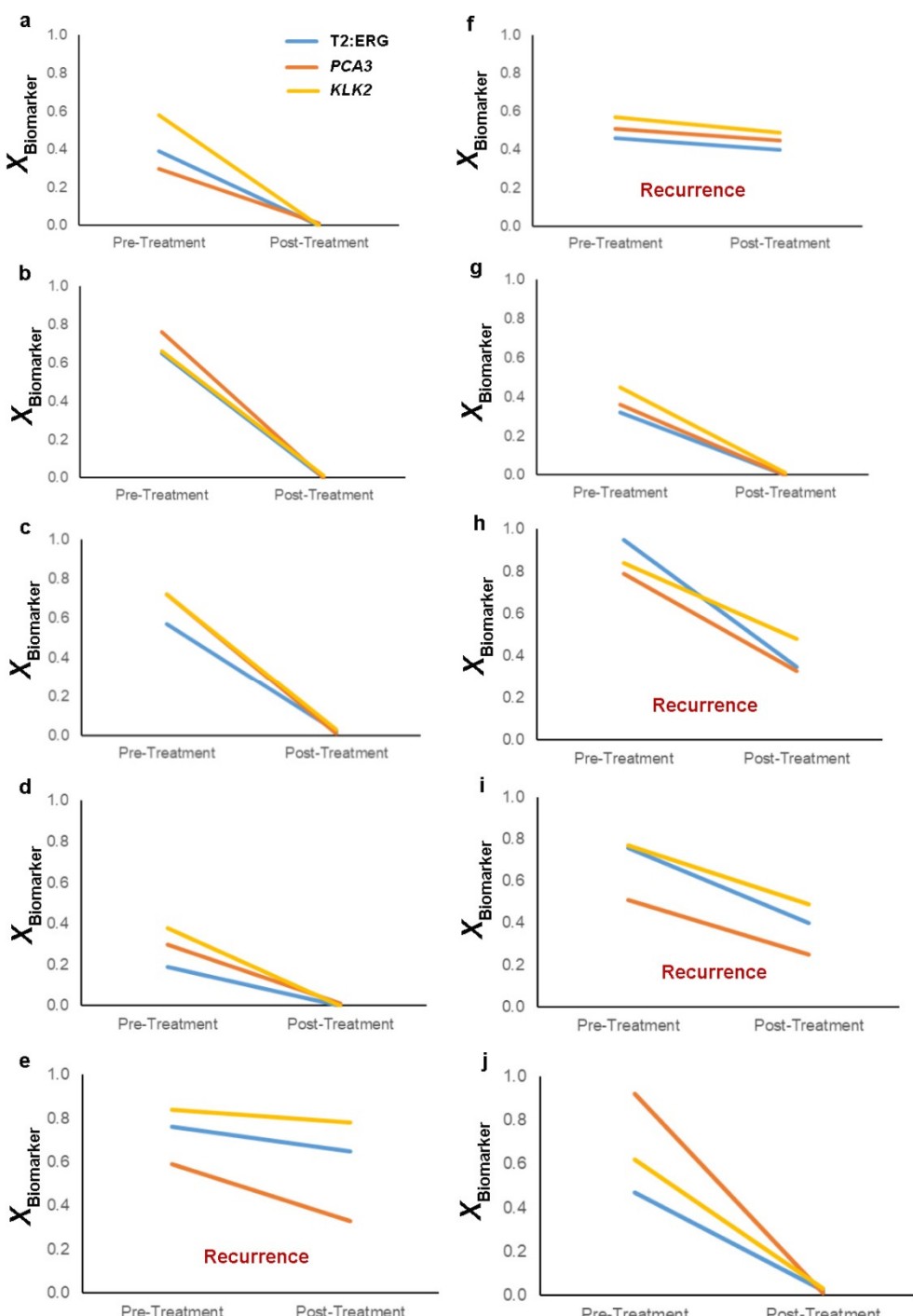

**Figure 4.** Non-invasive minimal residual disease monitoring in (**a**–**j**) pre- and post-treatment urinary liquid biopsy samples for 10 prostate cancer patients.

## 3. Materials and Methods

### 3.1. Materials

All reagents (Sigma-Aldrich, Castle Hill, Australia) were of analytical grade and used without further purification unless otherwise stated. UltraPure™ DNase/RNase-free distilled water (Invitrogen, Waltham, MA, USA) was used throughout the experiments. Primer sequences (Integrated DNA Technologies, Coralville, IA, USA) used in this work are shown in Table S1.

For PCa cell lines, cells were cultured in RPMI-1640 growth media (Life Technologies, Mulgrave, Australia) supplemented with 10% fetal bovine serum (Life Technologies, Mul-

grave, Australia) in a humidified incubator containing 5% $CO_2$ at 37 °C. For patient urine samples, ethics approval was obtained from The University of Queensland Institutional Human Research Ethics Committee (Approval No. 2004000047), and Royal Brisbane & Women's Hospital Human Research Ethics Committee (Ref No. 1995/088B). Informed consent was obtained from all subjects prior to sample collection, and methods pertaining to clinical samples were carried out in accordance with approved guidelines.

### 3.2. Magnetic Isolation of ctNAs

For direct magnetic enrichment of ctNA, 1 mL of urine sample was mixed with 250 µL of Dynabead SILANE magnetic beads (Invitrogen) suspended in binding buffer (100 mM Tris–HCl, pH 7.5; 10 mM EDTA, pH 8; 500 mM LiCl; 1% LiDS; 5 mM dithiothreitol). The mixture was mixed gently to resuspend magnetic beads and then incubated for 5 min at room temperature under continuous agitation. The magnetic beads were then magnetically washed twice with 1 mL of washing buffer (10 mM Tris–HCl, pH 7.5; 1 mM EDTA; 150 mM LiCl; 0.1% LiDS). The isolated ctNA were eluted from the magnetic bead surface by incubating washed magnetic beads in 15 µL of elution buffer (10 mM Tris–HCl, pH 7.5) at 70 °C for 2 min, immediately placing the mixture on a magnet, and collecting the eluted ctNA.

### 3.3. Isothermal Amplification of ctRNA Biomarkers on Magnetic Beads

Primer-functionalized magnetic beads were individually prepared for each ctRNA biomarker by incubating streptavidin-magnetic beads (1 µm, 100 µL) (Creative Diagnostics, New York, NY, USA) with respective biotin-modified forward primer sequences (Table S1) on a mixer(10 µM, 100 µL) at room temperature (30 min). After surface functionalization, excess unbound primers were removed using magnetic washing with Wash/Storage Buffer (Creative Diagnostics, USA), and primer-functionalized magnetic beads were resuspended in UltraPure DNase/RNase-free water (200 µL). For isothermal amplification of ctRNA biomarkers, the TwistAmp Basic RT-RPA kit (Twist-DX, Cambridge, MA, USA) was used with slight modifications to manufacturer's instructions. Eluted ctNA (1.5 µL), each reverse primer (375 nM) (Table S1), biotinylated uracil bases (20 nM) (Thermo Fisher Scientific, Australia) and associated set of forward-primer-functionalized magnetic beads (5 µL) were added to make a reaction volume (12.5 µL) prior to incubation (41 °C, 20 min).

### 3.4. Electrochemical Detection via Iron Oxide Nanozymes/Methylene Blue Bioelectrocatalytic System

After isothermal amplification on magnetic beads, amplicons for each ctRNA biomarker were magnetically washed with Wash/Storage Buffer (Creative Diagnostics, USA), incubated with streptavidin-modified iron oxide nanoparticles (10 nm, 1 µL) in phosphate buffer saline (10 mM, 0.5% triton-X pH 7.4) for 5 min to label biotinylated uracil bases, and washed again before resuspending in an electrolyte buffer (10 mM Tris, 100 mM KCl, 2.5 mM $MgCl_2$, 1 mM $CaCl_2$, pH 7.5) with 2 µM MB.

The resuspended mixture was dropped onto the gold working electrode surface of a DRP-C110-U75 screen-printed electrode (Metrohm, Herisau, Switzerland), and the iron-oxide-nanoparticle-attached amplicons were concentrated onto the working electrode surface by positioning a permanent magnet under the electrode. Chronoamperometry measurements were carried out using a CH1040C potentiostat (CH Instruments, Austin, TX, USA) at −450 mV, 30 s. All measurements were performed at room temperature.

### 3.5. Data Analysis

Normalized chronoamperometric signal (*X*) for each ctRNA biomarker was calculated with no-template background signal taken into consideration. For example:

$$X_{\text{T2:ERG}} = (\text{raw T2:ERG signal} - \text{no-template control background signal})/\text{no-template control background signal} \quad (1)$$

*3.6. qPCR Validation*

The KAPA SYBR® FAST One-Step qRT-PCR kit (KAPA Biosystems, Wilmington, MA, USA) was used to set up a single reaction volume for each sample (10 μL). Each reaction volume consisted of 1X KAPA SYBR® FAST qPCR Master Mix, each forward and reverse primer (200 nM) (Table S1), 1X KAPA RT Mix, ROX dye (50 nM) and eluted ctRNA (3 μL). RT-qPCR was performed using the Applied Biosystems® 7500 Real-Time PCR System (Thermo Fisher Scientific, Mulgrave, Australia). The cycling protocol was: 42 °C for 10 min to synthesize cDNA, followed by 95 °C for 5 min before cycling 40 times (95 °C for 30 s, 50 °C for 30 s and 72 °C for 1 min) and finished with 72 °C for 10 min.

**4. Conclusions**

An assay that leverages the multifunctional properties of streptavidin-modified iron oxide hybrid nanozymes was developed and demonstrated for MRD monitoring. Crucially, streptavidin-modified iron oxide hybrid nanozymes permitted rapid and cost-effective ctRNA analysis by magnetic enrichment of ctRNA biomarker amplicons on the electrode surface and facilitated bioelectrocatalytic signaling via leveraging amplicons as biological scaffolds. This rendered the assay approach more practical for PCa recurrence in urinary liquid biopsy samples by reducing analysis time and cost as compared to existing workflows. To build upon the outcome of this study, the next step could be for integration on a miniaturized microfluidic device to ensure sample preparation, and ctRNA biomarker amplification and detection can be performed together on the same platform instead of having separate workflows. This will facilitate the use of the assay towards automation or point-of-need usage beyond a laboratory environment. In overcoming the traditional challenges of ctNA analysis for MRD monitoring, it is anticipated that streptavidin-modified iron oxide hybrid nanozymes may essentially be used for broad ctNA detection applications.

**Supplementary Materials:** The following supporting information can be downloaded at: https://www.mdpi.com/article/10.3390/catal13010178/s1, Table S1: Primer sequences; Table S2: Validation of detection outcomes of assay against gold standard quantitative polymerase chain reaction (qPCR); Figure S1: Current measurements using only DNA-intercalating methylene blue (MB) or dual DNA-intercalating MB/iron oxide nanozymes bioelectrocatalytic cycling; Figure S2: Optimization of isothermal amplification times; Figure S3: Optimization of electrolyte buffer.

**Funding:** K.M.K. acknowledges funding from the Royal Brisbane and Women's Hospital (RBWH) Foundation.

**Conflicts of Interest:** K.M.K. is a scientist of XING Technologies Pty. Ltd., which has licensed intellectual property from The University of Queensland.

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
