# Peer review of "Multifunctional Hybrid Nanozymes for Magnetic Enrichment and Bioelectrocatalytic Sensing of Circulating Tumor RNA during Minimal Residual Disease Monitoring"

_catalysts, doi:10.3390/catal13010178_

Round 1

Reviewer 1 Report

The paper describes electrochemical sensor for the detection of circulating tumor DNAs enabling monitoring of tumor treatment effectiveness. The method is simple, non-invasive and sensitive. However, I am not sure that the paper fits the scope of Catalysts, because the role of nanozymes in the assay is not clear. In my opinion there is no appropriate controls demonstrating the conversion of leucomethylene blue (LB) to methylene blue (MB) by the iron oxide nanoparticles. It is curious how the author can use peroxidase-mimicking ability of iron oxide nanoparticles without the addition of peroxide substrate. The assay itself works well, but the mechanism is unclear. The author should demonstrate claimed (figure 1) principle of the assay using appropriate controls (proposed below) and additional experiments (e.g. conversion of LB to MB by the nanozymes in the absence of peroxide at neutral pH).

What is rationale to use iron oxide nanoparticles for signal enhancement? It seems that magnetism of microbeads is sufficient to catch DNA near the sensor surface (as it is sufficient to isolate beads from urine samples). Is it really necessary that streptavidin to be tagged with magnetic nanoparticles (In the Author’s recent paper microbeads performed well by itself: 10.1021/acssensors.0c01512)? It is important because iron oxide nanoparticles is not the most efficient nanozyme (see discussion in papers 10.1021/acsnano.2c02966 and 10.1021/acsnano.1c07520 as well as comparative study 10.1038/s41467-019-08657-5). I recommend authors repeat the experiment with cancer cells culture fluid (which is perhaps more available than urine samples) using HRP-streptavidin conjugate. Taking into account higher activity of HRP, lower LOD and higher analytical signal could be achieved. Moreover, in the previous paper (10.1021/acssensors.0c01512) more cheap oxidizing agent Fe(CN)63– was successfully applied and LOD was even lower – 5 copies of DNA.

pH optimization ­– iron oxide nanozymes are usually active at acidic pH (see e.g. Figs 17-20 in the SI of paper 10.1038/s41467-022-33098-y). Besides, H2O2 was not added. Please check the activity of IONP toward leucomethylene blue oxidation at neutral pH in the absence of H2O2. Was the impact of nanozymes’ catalytic activity in signal development really confirmed? Maybe nanoparticles just hinder the electron transfer? Please, perform the experiment without nanozymes and with streptavidin-tagged non-catalytic nanoparticles of comparable size, e.g. silica NPs. In the previous paper (10.1021/acssensors.0c01512) the pH of 7.5 was optimal however iron oxide nanoparticles were not used.

Result of analysis is expressed as a ratio (“X”) between sample and background with no positive control or calibrators included. This approach requires good day-to-day reproducibility of the assay, if comparison of “X” between samples obtained on different days is necessary. Have you measured day-to-day reproducibility of the assay (in terms of “X” value)?

Please add reference on the target gene expression levels in cell lines DuCap, LnCap, 22Rv1. Ref. 17 is the paper by the same author that claims “…well-characterized PC cell lines (DuCap, LnCap, 22Rv1) with known expression of the four target genes” and contains no confirmation of expression levels (e.g. PCR analysis, reference to earlier papers etc.).

Self-citations – 11 of 28 refs. Are all the refs are relevant?

Limitations of the method and possible solutions should be briefly discussed.

Lines 25-32 – remove template sentences

Line 39 – biomaRkers

Line 134 – “electrode surface damage lower pH” – please check the phrase

Line 153 – check spaces between words

Lines 170-171 – check typos

Figure 1-4 – please, make the text, labels and symbols larger

Author Response

Reviewer #1

The paper describes electrochemical sensor for the detection of circulating tumor DNAs enabling monitoring of tumor treatment effectiveness. The method is simple, non-invasive and sensitive. However, I am not sure that the paper fits the scope of Catalysts, because the role of nanozymes in the assay is not clear. In my opinion there is no appropriate controls demonstrating the conversion of leucomethylene blue (LB) to methylene blue (MB) by the iron oxide nanoparticles. It is curious how the author can use peroxidase-mimicking ability of iron oxide nanoparticles without the addition of peroxide substrate. The assay itself works well, but the mechanism is unclear. The author should demonstrate claimed (figure 1) principle of the assay using appropriate controls (proposed below) and additional experiments (e.g. conversion of LB to MB by the nanozymes in the absence of peroxide at neutral pH).

Reply: The reviewer’s feedback is appreciated.

  1. What is rationale to use iron oxide nanoparticles for signal enhancement? It seems that magnetism of microbeads is sufficient to catch DNA near the sensor surface (as it is sufficient to isolate beads from urine samples). Is it really necessary that streptavidin to be tagged with magnetic nanoparticles (In the Author’s recent paper microbeads performed well by itself: 10.1021/acssensors.0c01512)? It is important because iron oxide nanoparticles is not the most efficient nanozyme (see discussion in papers 10.1021/acsnano.2c02966 and 10.1021/acsnano.1c07520 as well as comparative study 10.1038/s41467-019-08657-5). I recommend authors repeat the experiment with cancer cells culture fluid (which is perhaps more available than urine samples) using HRP-streptavidin conjugate. Taking into account higher activity of HRP, lower LOD and higher analytical signal could be achieved. Moreover, in the previous paper (10.1021/acssensors.0c01512) more cheap oxidizing agent Fe(CN)63– was successfully applied and LOD was even lower – 5 copies of DNA.

Reply: The rationale of using the iron oxide nanoparticles is to exploit the dual magnetic and enzymatic functions of this hybrid nanomaterial for both magnetic enrichment and bioelectrocatalytic cycling of ctRNA biomarker amplicons. Moreover, as compared to magnetic beads, the nanosize of iron oxide nanoparticles is more compatible for use with NA molecules.  Main text (Page 2, Lines 71-73, 85-86) and manuscript title are modified to clarify this point.

Although it is agreed that iron oxide may not be the most efficient nanozyme as compared to biological HRP, it has the advantage of providing additional superparamagnetic property to bring the ctRNA biomarker amplicons closer to the magnetic bead surface and the electrode surface for more efficient bioelectrocatalysis. Thus, it is respectfully considered that comparison with HRP-streptavidin conjugates is beyond the scope of this concept. Instead, the signal enhancement due to bioelectrocatalytic cycling is now demonstrated with additional data (Figure S1) of chronoamperometric measurements of methylene blue in presence and absence of iron oxide nanozymes. Main text (Page 3, Lines 100-102) is now modified to describe this new dataset.

Regarding the lower LOD observed in the previous work, it is noteworthy to point out that a different set of biomarkers were being detected and the difference of five copies is likely due to the inherent difference in biomarker sequences affecting amplification efficiency.    

  1. pH optimization ­– iron oxide nanozymes are usually active at acidic pH (see e.g. Figs 17-20 in the SI of paper 10.1038/s41467-022-33098-y). Besides, H2O2 was not added. Please check the activity of IONP toward leucomethylene blue oxidation at neutral pH in the absence of H2O2. Was the impact of nanozymes’ catalytic activity in signal development really confirmed? Maybe nanoparticles just hinder the electron transfer? Please, perform the experiment without nanozymes and with streptavidin-tagged non-catalytic nanoparticles of comparable size, e.g. silica NPs. In the previous paper (10.1021/acssensors.0c01512) the pH of 7.5 was optimal however iron oxide nanoparticles were not used.

Reply: In the listed previous work, the bioelectrocatalytic system was different in the use of intercalating MB and diffusive Fe(CN)63- redox reporters. In this work, the bioelectrocatalytic system is one which utilized intercalating MB and amplicon-incorporated iron oxide nanozymes. It is known that iron oxide nanozymes can retain catalytic activity over a wide range of pH 1-12 (Singh, Front. Chem, 2012). Hence, the similar optimal pH of 7.5 in both studies is likely for keeping the amplicons stable as biological scaffolds on the magnetic bead surface, as well as the electrode surface intact (Page 4, Line 130), during bioelectrocatalytic cycling.

In addressment of the reviewer’s first comment, the impact of the iron oxide nanozymes in signal enhancement is now addressed with additional data (Figure S1) of chronoamperometric measurements of methylene blue in presence and absence of iron oxide nanozymes. Main text (Page 3, Lines 100-102) is now modified to describe this new dataset.

  1. Result of analysis is expressed as a ratio (“X”) between sample and background with no positive control or calibrators included. This approach requires good day-to-day reproducibility of the assay, if comparison of “X” between samples obtained on different days is necessary. Have you measured day-to-day reproducibility of the assay (in terms of “X” value)?

Reply: The day-to-day reproducibility is demonstrated with the three independent technical replicates during the evaluation of detection sensitivity (Figure 3). This is now clarified with main text modifications (Page 4, Lines 163-164).

  1. Please add reference on the target gene expression levels in cell lines DuCap, LnCap, 22Rv1. Ref. 17 is the paper by the same author that claims “…well-characterized PC cell lines (DuCap, LnCap, 22Rv1) with known expression of the four target genes” and contains no confirmation of expression levels (e.g. PCR analysis, reference to earlier papers etc.).

Reply: The references on expression level characterization by separate research groups have been added. Main text (Page 4, Line 143) is now modified

  1. Self-citations – 11 of 28 refs. Are all the refs are relevant?

Reply: These references are all relevant in showing the progress of ctRNA detection in liquid biopsies and bioelectrocatalytic cycling. Nonetheless, additional references of the latest relevant studies by separate research groups in the field have been added appropriately.  

  1. Limitations of the method and possible solutions should be briefly discussed.

Reply: The limitations of having separate workflows for sample preparation, ctRNA biomarker amplification and detection are now discussed. Main text (Page 8, Lines 276-280) is modified.

  1.  

Lines 25-32 – remove template sentences

Line 39 – biomaRkers

Line 134 – “electrode surface damage lower pH” – please check the phrase

Line 153 – check spaces between words

Lines 170-171 – check typos

Figure 1-4 – please, make the text, labels and symbols larger

Reply: The listed errors have been corrected.

Reviewer 2 Report

In this work, the author reported the use of streptavidin-modified iron oxide hybrid nanozymes for magnetic enrichment and bioelectrcatalytic sensing of three prostate cancer (PCa) ctRNA biomarkers with high detection specificity and sensitivity (10 copies) over an ultrabroad dynamic range (five orders of magnitude). However, there are some problems need to be solved before its publication.

1.     Line 59 on page 2, The description of the article is inconsistent with the contents of the references 18, the peroxidase-like activity of iron oxide nanoparticles was first discovered in 2007 in reference 18 but not during electrochemical measurements. In addition, since this work is related to nanozyme, the latest definition and perspective should be introduced, e.g. Nano Today, 2021, 40, 101269; Acc. Mater. Res. 2021, 2 (7), 534-547. Regarding to the biosensor application of nanozymes, surface modification may be matter. The authors should add some discussion on this aspect and some latest reports may be helpful, e.g. Exploration, 2021, 1 (1), 75-89.

2.     Why did the authors modify primers to magnetic beads? What is the function of magnetic beads in this system? Can its function be replaced by streptavidin-modified iron oxide? The author state that “This was probably due to maximal coverage of ctRNA biomarker amplicons on the magnetic bead surface at 20 min during magnetic enrichment on the electrode surface thus leading to no signal improvement beyond this timepoint”, Why not abandon the use of this bead or use smaller size beads with larger specific surface area?

3.     In 2.2, author performed the optimization of bioelectrocatalytic signaling process,according to reference “Biosensors and Bioelectronics, 2022:114739”, the amount of streptavidin on the nanozyme also needs to be optimized. The adsorption of more streptavidin on the nanozyme can make more nanozymes in the detection system and improve the detection sensitivity.

4.     The author state that “In particular, the close proximity of iron oxide nanozymes and MB labels within the ctRNA biomarker amplicons is beneficial for significantly enhacing the resultant electrochemical signals”, How about the assay sensitivity without nanozymes? This can directly prove the importance of nanozyme in this system.

5.     In 2.5, the author tested urine samples of 10 patients. Is there a clinical method for detecting ctRNA biomarker in urine? How does the results compare with the results of the authors?

6.     How about the assay sensitivity in this work compared to that in other work such as the use of duffusive redox labels or with the method used clinically.

7.     The language in the article needs polishing, please delete the writing instructions in line 25-33 on page 1. There is an additional thus in line 170 on page 5. Line 171 on page 5, nananozymes need to be nanozymes. Line 134 on Page 4, there need a “at” before “lower pH”.

Author Response

Reviewer #2

In this work, the author reported the use of streptavidin-modified iron oxide hybrid nanozymes for magnetic enrichment and bioelectrcatalytic sensing of three prostate cancer (PCa) ctRNA biomarkers with high detection specificity and sensitivity (10 copies) over an ultrabroad dynamic range (five orders of magnitude). However, there are some problems need to be solved before its publication.

Reply: The reviewer’s feedback is appreciated.

  1. Line 59 on page 2, The description of the article is inconsistent with the contents of the references 18, the peroxidase-like activity of iron oxide nanoparticles was first discovered in 2007 in reference 18 but not during electrochemical measurements. In addition, since this work is related to nanozyme, the latest definition and perspective should be introduced, e.g. Nano Today, 2021, 40, 101269; Acc. Mater. Res. 2021, 2 (7), 534-547. Regarding to the biosensor application of nanozymes, surface modification may be matter. The authors should add some discussion on this aspect and some latest reports may be helpful, e.g. Exploration, 2021, 1 (1), 75-89.

Reply: The mentioned sentence has been rephrased for accuracy and the suggested references are added. Main text (Page 2, Lines 48-50) is modified.

Surface modification may affect the peroxidase-mimicking catalytic activity of iron oxide nanozymes. However, it is noted that the streptavidin surface modification of iron oxide nanoparticles in this study did not remove the catalytic activity and is still effective for signal enhancement due to bioelectrocatalytic cycling, as demonstrated with additional data (Figure S1) of chronoamperometric measurements of methylene blue in presence and absence of iron oxide nanozymes. Main text (Page 3, Lines 100-102) is modified to discuss these aspects. 

  1. Why did the authors modify primers to magnetic beads? What is the function of magnetic beads in this system? Can its function be replaced by streptavidin-modified iron oxide? The author state that “This was probably due to maximal coverage of ctRNA biomarker amplicons on the magnetic bead surface at 20 min during magnetic enrichment on the electrode surface thus leading to no signal improvement beyond this timepoint”, Why not abandon the use of this bead or use smaller size beads with larger specific surface area?

Reply: The main function of the magnetic beads is to provide a solid substrate for specific isothermal amplification of ctRNA biomarkers. Thus, primers to specifically recognize the corresponding target biomarker are required. In contrast, by labeling the amplicons on the magnetic bead surface, the main dual functions of the streptavidin-modified iron oxide nanoparticles are to provide magnetic enrichment of the amplicons to the electrode surface and detection signal enhancement. Moreover, as compared to magnetic microbeads, the nanosize of iron oxide nanoparticles is more compatible for use with NA molecules.  Main text (Page 2, Lines 71-73, 85-86) and manuscript title are modified to clarify this point.

Smaller-sized magnetic beads are not ideal as larger surface area/magnetic bead allows higher accessibility of polymerase enzymes to nucleic acid strands on the magnetic bead surface during isothermal amplification.

  1. In 2.2, author performed the optimization of bioelectrocatalytic signaling process,according to reference “Biosensors and Bioelectronics, 2022:114739”, the amount of streptavidin on the nanozyme also needs to be optimized. The adsorption of more streptavidin on the nanozyme can make more nanozymes in the detection system and improve the detection sensitivity.

Reply: Due to biotin-streptavidin interactions, the quantity of streptavidin-modified iron oxide nanozymes for generating the detection signal is essentially limited by the number of biotins within the amplicons on the magnetic bead surface. Thus, the isothermal amplification time was optimized to generate maximal number of amplicons on the magnetic bead surface (Figure S2).    

  1. The author state that “In particular, the close proximity of iron oxide nanozymes and MB labels within the ctRNA biomarker amplicons is beneficial for significantly enhacing the resultant electrochemical signals”, How about the assay sensitivity without nanozymes? This can directly prove the importance of nanozyme in this system.

Reply: The impact of the iron oxide nanozymes in signal enhancement is now addressed with additional data (Figure S1) of chronoamperometric measurements of MB in presence and absence of iron oxide nanozymes. Main text (Page 3, Lines 100-102) is now modified to describe this new dataset.

  1. In 2.5, the author tested urine samples of 10 patients. Is there a clinical method for detecting ctRNA biomarker in urine? How does the results compare with the results of the authors?

Reply: The gold standard method for targeted ctRNA detection in urine is quantitative polymerase chain reaction (qPCR). Gold standard qPCR was unable to be performed on the clinical urine specimens due to insufficient quantity. However, qPCR was performed on contrived urine samples with different prostate cancer cell line RNA. The qPCR results (Table S2) were found to be 100% concordant with the assay results shown in this study (Figure 2). Main text (Page 4, Lines 143-145) is now modified to describe this new dataset.

  1. How about the assay sensitivity in this work compared to that in other work such as the use of duffusive redox labels or with the method used clinically.

Reply: Main text (Page 4, Lines 170-172) is now modified to discuss the achieved detection sensitivity in this study in comparison with the use of diffusive redox labels and gold standard qPCR.

  1. The language in the article needs polishing, please delete the writing instructions in line 25-33 on page 1. There is an additional thus in line 170 on page 5. Line 171 on page 5, nananozymes need to be nanozymes. Line 134 on Page 4, there need a “at” before “lower pH”.

Reply: The listed errors have been corrected.

Round 2

Reviewer 1 Report

1. The phrase “compared to magnetic beads, the 85 nanosize of iron oxide nanoparticles is more compatible for use with NA molecules” (Line 86) is unclear. More compatible by what means? Do magnetic beads (in contrast to nanoparticles) interact with NA in somewhat destructive way? How did author measure this “compatibility”? If magnetic beads have some negative impact on NA, why are they used for NA isolation and present on the electrode surface together with nanoparticles in the course of the assay? In my opinion, clarification is necessary here.

2. The author demonstrated benefits of IONP in terms of analytical signal increase. Nevertheless, I have some doubt about catalytic mechanism of iron oxide nanoparticles. The author replied “iron oxide nanozymes can retain catalytic activity over a wide range of pH 1-12 (Singh, Front. Chem, 2012)” . First of all, Frontiers in Chemistry published its first papers in 2013, therefore paper “Singh, Front. Chem, 2012” cannot exist. I think the author meant paper Singh S (2019) Nanomaterials Exhibiting Enzyme-Like Properties (Nanozymes): Current Advances and Future Perspectives. Front. Chem. 7:46. doi: 10.3389/fchem.2019.00046. In this review there is no information about peroxidase-like activity of iron oxide nanoparticles at neutral pH. The phrase from this review “nanoparticles remain stable and retain their catalytic activity after the incubation at a broader range temperature [4–90°C and pH (1–12)” references to article 10.1038/nnano.2007.260, in which nanoparticles were first incubated at various pH, and then their remaining activity was measured at ACIDIC pH (pH 3.5). Moreover, this very review cites paper by Chen, Z., et al., 10.1021/nn300291r in which the absence of any peroxidase-like activity of iron oxide nanoparticles at physiological pH (7.4) was clearly demonstrated (fig. 4B). Therefore, I find arguments of the author regarding catalytic activity of nanozymes at neutral pH unconvincing. Catalytic activity of nanozymes in the assay must be proven. The author showed that nanoparticles enhance the signal, but claimed mechanism needs experimental verification. In its present form, conclusions are not supported by the data. Is it possible, that magnetic effect of nanoparticles (they get NA closer to electrode’s surface) is solely responsible for signal enhancement?

3. Please, enlarge axis labels in figures 2, 3, and 4.
